# A Novel Piezo Inertia Actuator Utilizing the Transverse Motion of Two Parallel Leaf-Springs

**DOI:** 10.3390/mi14050954

**Published:** 2023-04-27

**Authors:** Pingping Sun, Zhike Xu, Long Jin, Xingxing Zhu

**Affiliations:** 1School of Physics and Information Engineering, Jiangsu Second Normal University, Nanjing 211200, China; 2School of Electrical Engineering, Southeast University, Nanjing 210096, China; zhuxingxing@seu.edu.cn (X.Z.); jinlong@seu.edu.cn (L.J.)

**Keywords:** piezo inertia actuator, stick-slip, leaf-spring, flexure hinge

## Abstract

A novel linear piezo inertia actuator based on the transverse motion principle is proposed. Under the action of the transverse motion of two parallel leaf-springs, the designed piezo inertia actuator can achieve great stroke movements at a fairly high speed. The presented actuator includes a rectangle flexure hinge mechanism (RFHM) with two parallel leaf-springs, a piezo-stack, a base, and a stage. The mechanism construction and operating principle of the piezo inertia actuator are discussed, respectively. To obtain the proper geometry of the RFHM, we have used a commercial finite element program COMSOL. To investigate the output characteristics of the actuator, the relevant experiment tests including loading capacity, voltage characteristic, and frequency characteristic are adopted. The maximum movement speed and the minimum step size are 27.077 mm/s and 32.5 nm, respectively, confirming that the RFHM with two parallel leaf-springs can be used to design a piezo inertia actuator with a high speed and accuracy. Therefore, this actuator can be used in applications with fast positioning and high accuracy.

## 1. Introduction

Piezo inertia actuators utilize the inertia of a mover or stage to enable small displacements employing uninterrupted friction contact, thereby achieving precise and accurate positioning. Since the invention of the first piezo inertia actuators was developed by Pohl [1,2], a number of piezo inertia actuators have been proposed by developers across the world. Given the various advantages of piezo inertia actuators, such as those that are lightweight, a compact structure, fine position, and fast response, they are widely and successfully used in a variety of industries, such as zooming and image stabilization in camera modules [3], micro-positioner for microscopy applications [4,5,6,7,8,9,10,11,12,13], and micro robot applications [14].

In recent years, a new type of piezo inertia actuator based on the parasitic motion principle was presented. Through the combination of a piezo-stack and a flexure hinge mechanism (FHM), the piezo-stack makes the FHM synchronously deform in the vertical and transverse directions. The vertical deformation makes the preload force and corresponding frictional force between the stage and friction piece increase and decreases in turn during operation. The transverse deformation is utilized to drive the stage to move forward or backward. Based on the above principle, several researchers developed piezo inertia actuators in a multitude of designs, such as Li et al. [15], who presented a parallelogram-type piezo inertia actuator with a FHM, with the vertical and transverse displacements of 10 μm and 15.95 μm, respectively, a displacement ratio of 62.7%, and a maximum velocity of 14.25 mm/s. Li et al. [16] presented a lever-type piezo inertia actuator with a FHM, with the vertical and transverse displacements of 10 μm and 20.67 μm, respectively, and a displacement ratio of 48.3%. The designed maximum speed is 7.12 mm/s. Xu et al. [17,18] presented a feasibility foot actuator with a high speed of 18.37 mm/s under the action of the transverse displacement of the FHM. The transverse displacement is 27.27 μm. Zhang et al. [19] presented a triangular-compliant driving mechanism utilizing the parasitic motion of the triangular with a stable velocity of 0.7 mm/s. The vertical and transverse displacements are 11 μm and 22 μm, respectively, and the displacement ratio is 50%. Liu et al. [20] presented an inertial piezo actuator with double stators, utilizing the lever and flexible driving beams. The transverse displacement is 28.6 μm, and the maximum speed of the actuator is only 1042.99 mrad/s. Huang et al. [21] presented a piezoelectric actuator using a lever mechanism with an output speed of 1.4 mm/s and a transverse displacement of 14.23 μm. Koc et al. [22] presented a two-phase inertial drive motor. It is worth noting that the inertial motor does not use a flexible hinge amplification mechanism to amplify the piezo-stack displacement. Through the coordination of the two piezo-stacks, the hysteresis characteristics of their respective piezo-stacks are offset by each other. The maximum speed is 16 mm/s. Many previous piezo inertia actuators have achieved a greater transverse displacement, but their vertical displacement components are great, which restricts the speed of piezo inertia actuators to a certain degree.

Here, the displacement ratio is defined as the radio of the vertical displacement component to the transverse displacement component, which represents the proportion of vertical displacement. The lower the displacement ratio is, the higher the proportion of transverse displacement generation. In other words, when the output displacement generated by the piezo-stack can be fully used to increase the transverse displacement component of the FHM, the stage will achieve a fairly high speed. The moving speed of the stage can be expressed by Equation (1):(1)V=f×ΔL
where V denotes the speed of the stage; f denotes the frequency of the driving voltage; and ΔL is the magnitude of step displacement of the stage. It is assumed that the driving frequency is constant, and the transverse displacement component of the FHM is approximately equal to the step displacement ΔL. According to Equation (1), it can be noted that the speed of the stage increases as the transverse displacement component increases.

To reduce the vertical displacement component and increase the transverse displacement component, the study proposes a novel piezo inertia actuator driven primarily by the transverse motion of the RFHM with two parallel leaf-springs. Through the cooperation of one pair of the parallel leaf-springs, the piezo-stack’s elongation displacement deforms a minor vertical displacement and a major transverse displacement of the flexure hinge mechanism. In this way, great stroke movements with a fairly high speed at the stage can be achieved.

In addition, the statics model of a single leaf-spring and the RFHM are discussed, respectively, and a commercial finite element program COMSOL is used to study the characteristic of the RFHM. The operating principle is verified by the results of the relevant tests.

## 2. Design and Analysis

### 2.1. Construction of Actuator

Figure 1 shows the mechanical construction of the linear piezo inertia actuator. It is composed of a rectangle flexure hinge mechanism (RFHM) with two parallel leaf-springs, a piezo-stack, a base, and a stage. The piezo-stack is 10 mm in length, nested in the RFHM by mechanical capture and placed at an angle to the centerline of mechanism, and it reaches a stroke of approximately 16 μm with 100 V and consequently deforms the RFHM. Two wedges are used to adjust the preload force applied to the piezo-stack. As the maximum output force of the selected piezo-stack is 200 N, the optimal preload applied to the piezo-stack is about 10% to 15% of the maximum output force, and the corresponding value is about 20 N to 30 N. The stage is used as the mover, with a small coefficient of friction motion and only one translational degree of freedom. Two mounting screws engage the RFHM to attach the mechanism to the base firmly. Moreover, the preload screw is exposed to the bottom of the base, which is utilized to adjust the preload force between the friction head and the stage. The friction head is bonded to the RFHM. ZrO_2_ is chosen for the friction head, which has smooth dry hard surfaces. Al7075 is used for the RFHM, which has excellent stretchy characteristic. The piezo-stack is PZT-5H (from NEC AE0203D08H09DF). The stage is made of stainless steel S6205, which has a smooth and strong surface. Therefore, stainless steel S6205 and ceramic ZrO_2_ form a friction pair, with a low friction coefficient and wear resistance. The relevant material constants are shown in Table 1.

### 2.2. Mechanical Model of RHFM

Two flexure hinges in series form a leaf-spring, as shown in Figure 2a. The flexure hinges of leaf-spring are made slender and thin. The structure parameters of a single leaf-spring are listed in Table 2. As the top of the leaf-spring is subjected to only one transverse force ${\;f}_{x}$, it will deform the leaf-spring both in *x*-direction and y-direction, respectively, being, namely, $u_{x}$ and $\;u_{y}$. The stiffness of a single leaf-spring in the x-direction and y-direction can be expressed by Equations (2) and (3), respectively.
(2)$$C_{x}=\frac{20Eba^{3}a_{1}^{3}}{(8L_{1}^{3}-12LL_{1}^{2}+6L^{2}L_{1})(a_{1}^{3}-a^{3})+a^{5}}$$
(3)$$C_{y}=\frac{3Eaa_{1}b}{2L_{1}a_{1}-2L_{1}a+aL}$$

Here, *E* denotes Young’s modulus of Al7075, and *b* denotes the thickness of the leaf-spring.

The displacement $u_{x}$ can be expressed as follows, where $u_{x}$ denotes the elastic deformation of the top of a single leaf-spring in *x*-direction:(4)$$u_{x}=\frac{f_{x}}{C_{x}}$$

However, no force in y-direction is applied to the top of a single leaf-spring. Thus, the y-direction displacement is coupled to the *x*-direction. The displacement $u_{y}$ can be mathematically expressed by the following equation:(5)$$u_{y}=\frac{u_{x}^{2}}{2(L-L_{1})}$$

The designed flexure hinge mechanism consists of two leaf-springs and two beams to form a whole, as shown in Figure 2b. One pair of the parallel leaf-springs can be together used for friction free motion. Through a double parallel leaf-spring, the parasitic displacement $u_{y}$ can be neglected to a large extent. That is to say, the piezo-stack’s axial direction displacement can be almost used to deform the RFHM’s displacement in the *x*-direction, namely, $u_{x}$. The displacement $u_{x}$ is the transverse motion of the RFHM, which is utilized to drive the stage to run in the x-direction. The stiffness of the RFHM in the *x*-direction and y-direction can be expressed by Equations (6) and (7), respectively.
(6)$$C_{x}{}^{\prime }=\frac{40Eba^{3}a_{1}^{3}}{(8L_{1}^{3}-12LL_{1}^{2}+6L^{2}L_{1})(a_{1}^{3}-a^{3})+a^{5}}$$
(7)$$C_{y}{}^{\prime }=\frac{6Eaa_{1}b}{2L_{1}a_{1}-2L_{1}a+aL}$$

The push force F generated from piezo-stack can be divided into two components as $F_{x}$ and $F_{y}$.


(8)
$$\left[\begin{array}{c} F_{x}\\ F_{y} \end{array}\right]=\left[\begin{array}{c} \sin\theta \\ \cos\theta \end{array}\right]F$$


Here, the angle θ is $10^{{}^{\circ}}$ and the push force F is 200 N. The push force is a product of the maximum output displacement and stiffness (specification: maximum displacement: 9.1±1.5 μm, stiffness: 22 N/μm, and generated force: 200 N).

Combining the above, Formulas (6)–(8), the displacement $u_{x}$ and $u_{y}$ of the top of the RHFM (friction head) can be expressed by Equations (9) and (10), respectively.
(9)$$U_{x}=\frac{F_{x}}{C_{x}{}^{\prime }}$$
(10)$$U_{y}=\frac{F_{y}}{C_{y}{}^{\prime }}-\frac{U_{x}^{2}}{2(L-L_{1})}$$

### 2.3. Simulation of RFHM

The static analysis model of the designed RFHM can be established by a commercial finite element program COMSOL. The material constants and the structure parameters of the leaf-spring used in the simulation are, respectively, listed in Table 2 and Table 3. Figure 3 shows the simulation results of the RFHM. The bottom surface of the RFHM is absolutely fixed during the simulation. With a driving voltage of 100 V, the simulation displacements of the friction head in the *x*-direction and y-direction are about 48.46 μm and 1.23 μm, respectively. The theoretical displacements of the friction head in the x-direction and y-direction can be calculated from Equations (6)–(10), which are about 48.05 μm and 1.39 μm, respectively, and are nearly equal to the simulation displacement results. Thus, the y-direction displacement can be ignored, compared with the great *x*-direction displacement.

### 2.4. Operating Principle

Typically, the inertia actuators are driven with a saw-tooth voltage signal with a linear slow rise and a sharp drop. The operation process of the piezo inertia actuator in a working circle is shown in Figure 4. There are three steps in a working circle.

Step 1: As shown in Figure 4a, firstly, as the piezo-stack has no power excitation at time t_0_, the rectangle flexure hinge mechanism and the stage remain practically stationary.

Step 2: From time t_0_ to t_1_, the piezo-stack slowly obtains power excitation with the linear rise of voltage signal. Simultaneously, the generated push force of piezo-stack will slowly deform the RFHM both in the x-direction and y-direction. The stage will move forward a step displacement ΔS by static frictional force between the stage and the friction head, as shown in Figure 4b. This is the so-called “stick” process.

Step 3: As shown in Figure 4c, the piezo-stack quickly contracts to its original position with a sharp drop of voltage signal. Simultaneously, the RFHM will quickly follow the piezo-stack closely. However, under the action of the inertia of the stage, the stage will stay nearly at the same position as shown in Figure 4b. Actually, the stage will move backward by a small displacement $\Delta S_{1}$ due to the sliding friction force. This is the so-called “slip” process, which lasts from time t_1_ to t_2_.

Therefore, the entire process is the so-called “stick-slip” mode, and the magnitude of step displacement ΔL of the stage can be obtained as follows:(11)$$\Delta L=\Delta S-\Delta S_{1}$$

For the reverse motion, the inertia actuator is driven by a saw-tooth voltage signal with a sharp rise and linear slow drop. The entire process is reversed to the “slip-stick” mode. By repeating these two modes, positive and negative continuous stepping motion can be achieved.

Under a certain high frequency, the “stick” processes in two modes are no more applicable, and the entire processes all turn into “slip-slip” modes. By repeating a “slip-slip” mode, the high-speed stage can be achieved.

## 3. Experiments and Results

### 3.1. Experimental System

To study the mechanical characteristics of the designed piezo inertia actuator, we set up a series of experiments. As shown in Figure 5a, the experimental system is composed of a computer, an amplifier, an oscilloscope, a signal generator, a laser sensor, and a controller. The saw-tooth voltage signal is produced by a signal generator (SDG1005; Siglent Technologies Co., Ltd., Shenzhen, China) and amplified by a power amplifier (E-501.00; Physik Instrumente Co., Ltd., Karlsruhe, Germany). An oscilloscope (ADS1042C; Atten Technologies Co., Ltd., Shenzhen, China) is used to monitor the change in the driving saw-tooth voltage signal. A laser sensor with a repeat accuracy of 20 nm and a controller with optional resolution of 1 nm (LK-H020/LK-G5001P; Keyence Co., Osaka, Japan) are used to measure the displacement and velocity of the stage. The actuator prototype is shown in Figure 5b, with dimensions of 60 × 60 × 13.5 mm^3^, machined by slow-feeding electrical discharge machining.

### 3.2. Transverse and Vertical Displacement Characteristics

To experimentally measure the transverse and vertical displacement of the RFHM, we established the experimental system for displacement measurement, as shown in Figure 6. The laser sensors 1 and 2 (LK-H020/LK-H150; Keyence Co., Osaka, Japan) are used to, respectively, measure the vertical and transverse displacement of the RFHM. Under the driving voltage of 100 V from the DC power (E05.A3, Harbin Core Tomorrow Science & Technology Co., Ltd., Harbin, China), the test results are listed in Table 3. It is noted that the vertical displacement $u_{y}$ is quite small relative to the transverse displacement $u_{x}$. Accordingly, the experiment results are very close to the results obtained by theoretical analysis and the finite element method. The slight deviation between the experimental value and the theoretical analysis value is mainly caused by the dimensional error of the RFHM during machining.

### 3.3. Frequency Characteristics

The minimum stepping displacement, also known as the resolution, is an important performance of the designed piezo inertia actuator. Therefore, it determines the precise positioning capability of an inertia actuator. To measure the minimum stepping displacement, we set the driving voltage to 10 V. Figure 7a shows the output displacements in four steps under the driving frequency of 1 Hz, and the cumulate output displacement of stage is 0.13 μm. So, the minimum stepping displacement is 32.5 nm.

However, a higher resolution has also been used in many practical applications. Therefore, the driving frequency of the actuator should be selected in the test. Figure 7b,c show the output displacement characteristics within 4 s at 100 V at the driving frequency range from 10 Hz to 100 Hz. The output displacement curve shows a linear increasing tendency. When the driving frequency increases, the output displacement and speed of the stage increase. It is worth noting that the step motion can be easily observed at 10 Hz and 20 Hz. The motion operation agrees well with the analysis procedure illustrated in Figure 4. The stepping motion is no more obvious above 30 Hz, and the motion turns into a “slip-slip” mode. Therefore, the stable motion of the actuator can be obtained in the slip–slip process. Under the driving frequency of 100 Hz, the cumulative output displacement of the stage is 12,268.67 μm within 4 s. Therefore, the effective stepping displacement is 30.67 μm.

Figure 8 shows the changes of the speed with the driving frequency. It is noted that the speed curve has three inflection points at 350 Hz, 550 Hz, and 900 Hz. At all the inflection points, the speed drops quickly. The quick decrease in velocity may be due to the vicinity of the frequencies of all the inflection points to the resonant frequencies of the different modes of the RFHM and the piezo-stack. Except for the inflection point of the speed curve, the speed increases with the frequency up to 800 Hz, and, later, the speed decreases. The maximum speed is 27.077 mm/s at 800 Hz and 100 V.

### 3.4. Voltage Characteristics

Section 3.3 discusses the changes of the output characteristics under various driving frequencies. The stepping motion obtained at 20 Hz is easily observed in the “stick–slip” operation mode. The step-like motion is no longer noticeable above 20 Hz, and the motion turns into a “slip–slip” operation mode. In this section, to further explore the changes of the output characteristics with various driving voltages, the driving frequencies of 20 Hz and 800 Hz are chosen in the subsequent experiments. Therefore, for clarity, the selected driving frequencies of 20 Hz and 800 Hz are discussed separately in the following experiments.

Figure 9a and Figure 10a present the output displacement and corresponding speed characteristics within 4 s at the driving frequency of 20 Hz, respectively. The step displacement has good linearity under different driving voltages. As the driving voltage increases from 50 V to 100 V, the output displacement increases from 46.02 μm to 411.02 μm, and the speed increases from 11.51 μm/s to 102.76 μm/s. However, the output displacement and the corresponding speed have a trend of slow increase below 70 V, and the continuous output characteristics are very unstable below 50 V.

Similarly, the output displacement and the corresponding speed characteristics within 4 s at the driving frequency of 800 Hz are illustrated in Figure 9b and Figure 10b, respectively. As the travel range of the stage is limited to 30 mm, the sampling time of output characteristics at 800 Hz is set to 1 s during the testing process. With an increase in driving voltage from 50 V to 100 V, the output displacement increases from 12.053 mm to 27.077 mm, and the speed increases from 12.05 mm/s to 27.077 mm/s. It is noted that the displacement curves have a very good linearity under two driving voltages of 90 V and 100 V. This means that the actuator has more stable motion in practice.

### 3.5. Load Characteristics

For the practical application, the loading capacity is one of the most essential parameters. For the application of fast focusing imaging in the white light interferometer, the lens with weight of fifty grams is installed on the actuator as the external load, and the required displacement resolution is 0.5 microns. Therefore, it is necessary to explore the load capacity of the designed actuator. Figure 11a shows the experimental system for measuring the load capacity of the actuator. Figure 11b illustrates the relationship between the external load and the speed at 800 Hz and 100 V. With an increase in external load form 0 g to 200 g, the speed of the stage decreases from 27.077 mm/s to 2.334 mm/s. When the external load is 50 g, the speed of the stage is 24.323 mm/s. This meets the detection requirements of the white light interferometer. The maximum external load of the actuator is about 200 g.

## 4. Comparison and Discussion

Table 4 compares some previous piezo inertia actuators with the designed actuator in this study in terms of maximum speed, resolution, load, driving frequency, voltage, transverse displacement, vertical displacement, and displacement ratio. It is noted that the speed and the displacement ratio of an actuator prototype have been significantly improved, demonstrating that the RFHM with two parallel leaf-springs achieves a higher speed, a minor vertical displacement, and a major transverse displacement. The resolution of 32.5 nm and the load of 200 g are also competitive in practical application.

## 5. Conclusions

In summary, a novel linear piezo inertia actuator driven by the transverse motion of the RFHM is proposed and manufactured in this study. Through the cooperation of one pair of the parallel leaf-springs, the RFHM has a minor vertical displacement and a major transverse displacement. The test results of the RFHM are close to those obtained by theoretic analysis and the finite element method. Thus, the correctness of the design of the RFHM is verified. The maximum speed is 27.077 mm/s under the driving frequency of 800 Hz at 100 V. The minimum stepping displacement is 32.5 nm under the driving frequency of 1 Hz at 10 V. Therefore, high-resolution step motion and large stroke movement with fairly high speed of the actuator can be achieved. This study confirms that the design of the RFHM with two parallel leaf-springs may offer a new idea for the design of a piezo inertia actuator with merits of high resolution, great stroke, and high speed.

## Figures and Tables

**Figure 1 micromachines-14-00954-f001:**
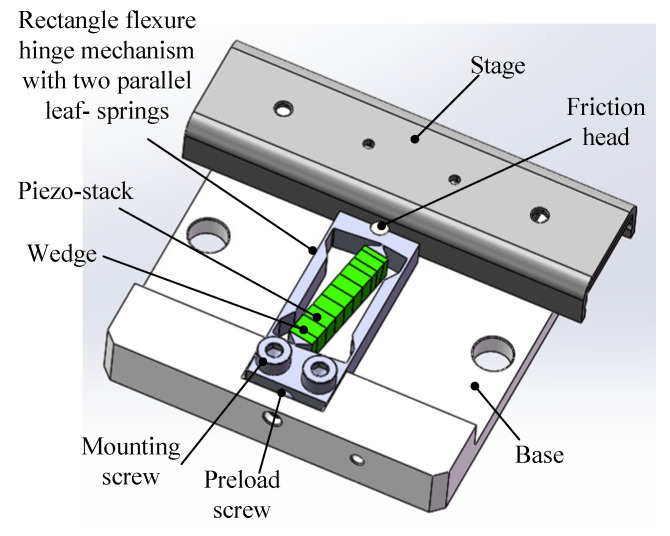
Construction of the designed piezo inertia actuator.

**Figure 2 micromachines-14-00954-f002:**
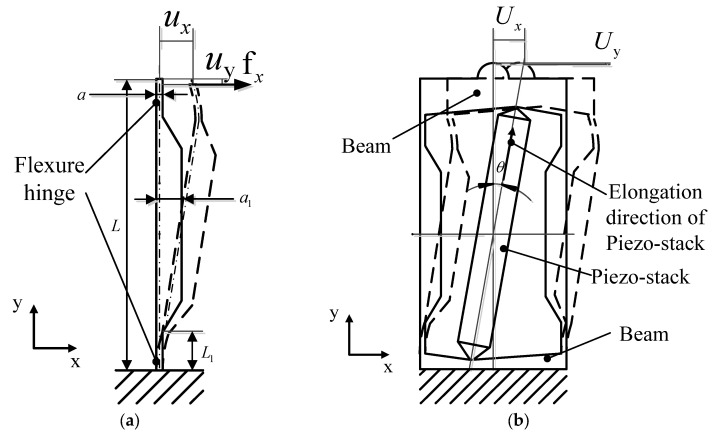
Rectangle flexure hinge mechanism: (**a**) structure of a single leaf-spring; (**b**) two leaf-springs in parallel.

**Figure 3 micromachines-14-00954-f003:**
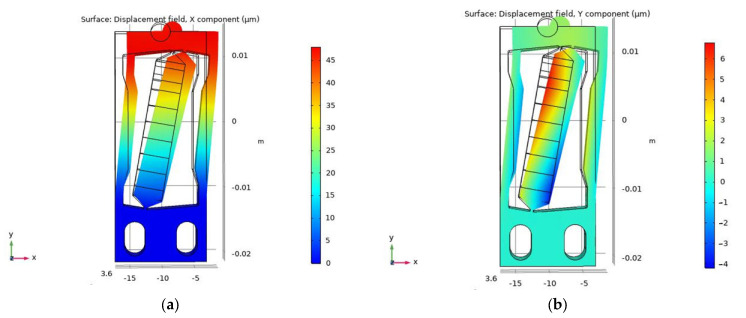
Static analysis simulation of the designed RFHM: (**a**) X-direction displacement; (**b**) Y-direction displacement.

**Figure 4 micromachines-14-00954-f004:**
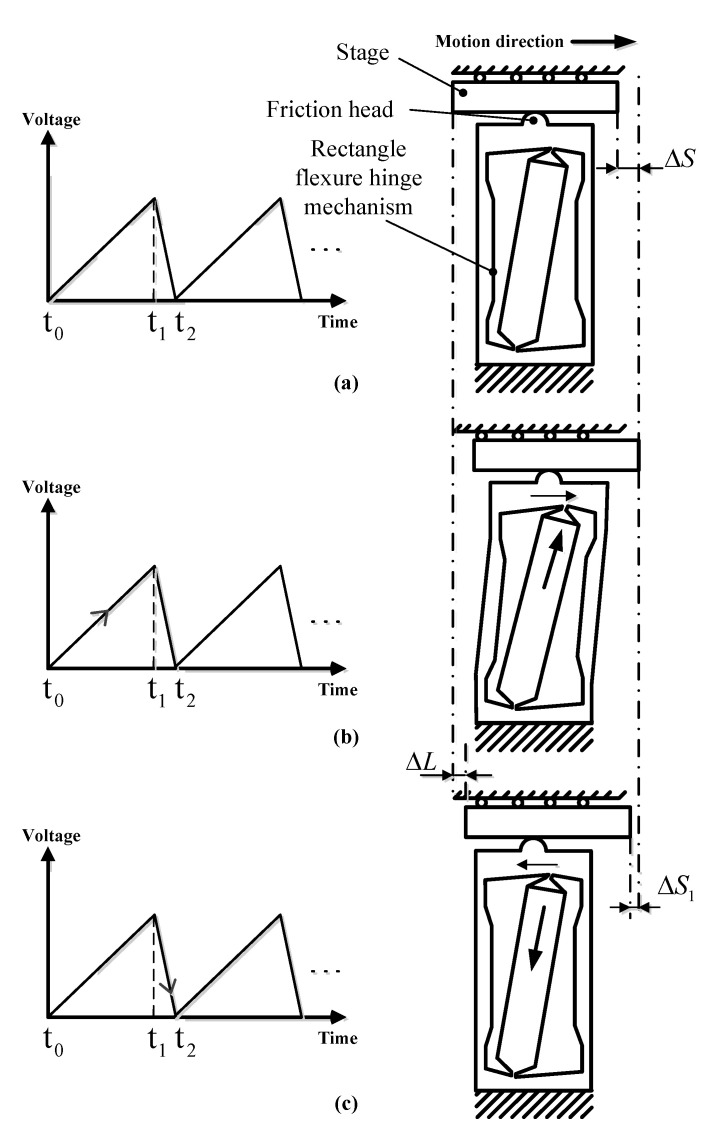
Operation process of the piezo inertia actuator in a working circle: (**a**) Step 1; (**b**) Step 2; (**c**) Step 3.

**Figure 5 micromachines-14-00954-f005:**
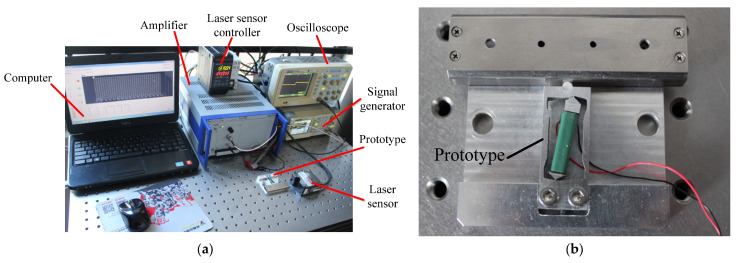
Experimental system of the designed actuator: (**a**) experimental system; (**b**) actuator prototype.

**Figure 6 micromachines-14-00954-f006:**
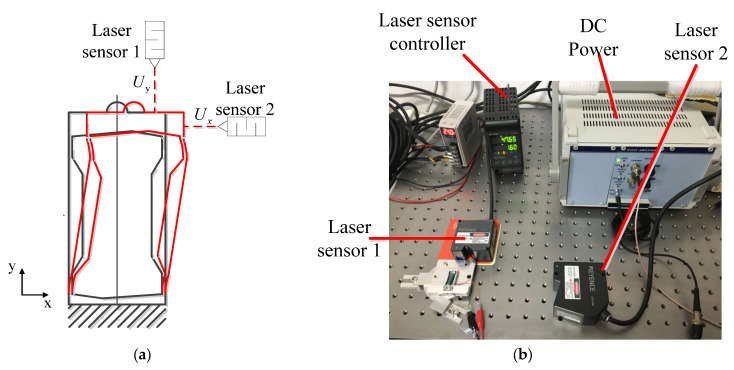
Displacement measurement experimental system of the RFHM: (**a**) schematic diagram of the experiment; (**b**) measurement system of the experiment.

**Figure 7 micromachines-14-00954-f007:**
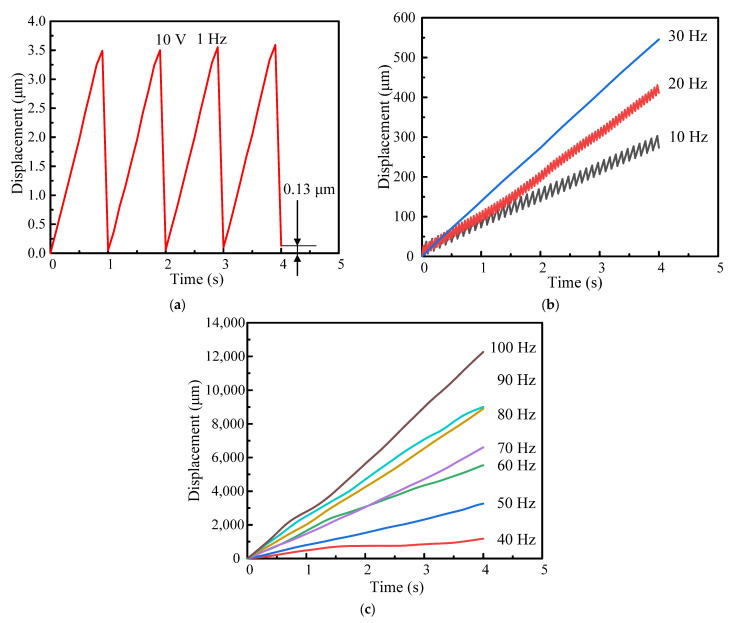
Output displacement characteristics with the driving frequency range from 1 to 100 Hz: (**a**) output displacements in four steps under the driving frequency of 1Hz; (**b**) the driving frequency range from 10 to 30 Hz; (**c**) the driving frequency range from 40 to 100 Hz.

**Figure 8 micromachines-14-00954-f008:**
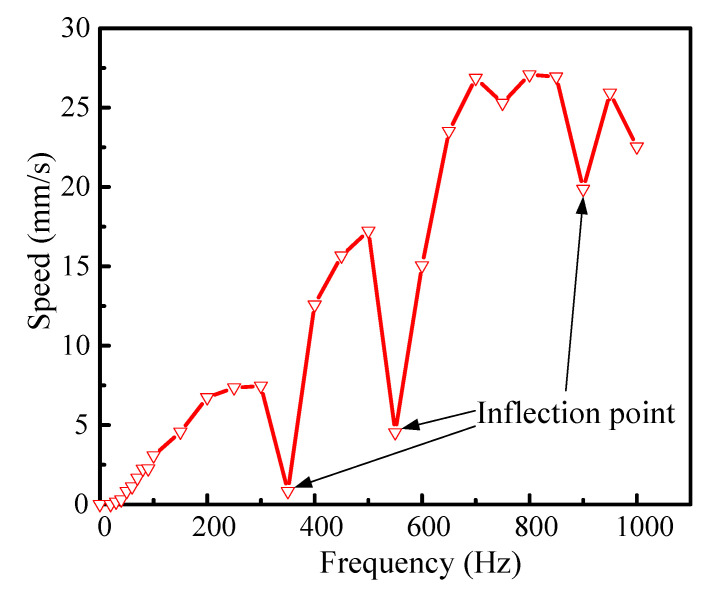
Speed-frequency characteristic curve.

**Figure 9 micromachines-14-00954-f009:**
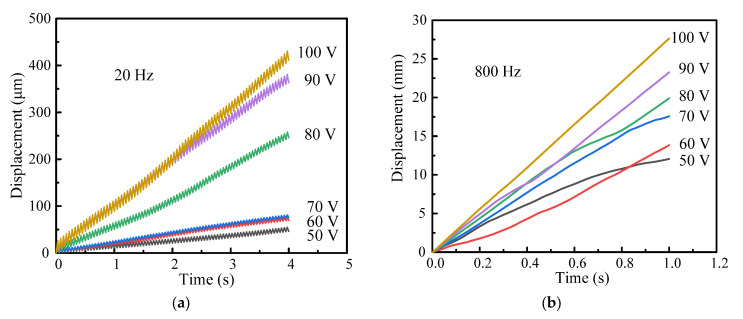
Output displacement characteristics with the driving voltage range from 50 to 100 V: (**a**) driving frequency of 20 Hz; (**b**) driving frequency of 800 Hz.

**Figure 10 micromachines-14-00954-f010:**
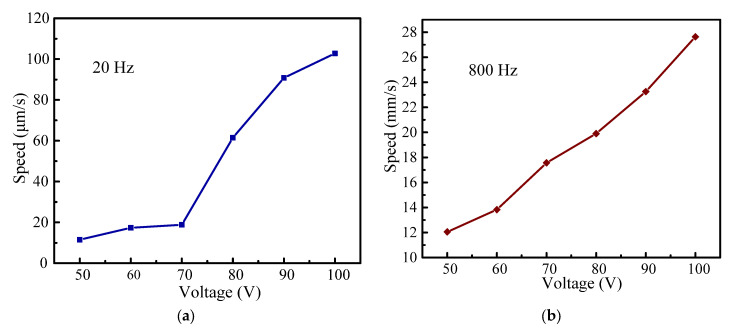
Output speed characteristics with the driving voltage range from 50 to 100 V: (**a**) driving frequency of 20 Hz; (**b**) driving frequency of 800 Hz.

**Figure 11 micromachines-14-00954-f011:**
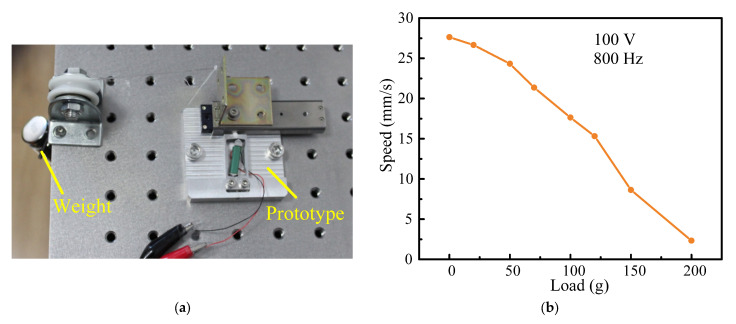
Load characteristics: (**a**) experimental system; (**b**) speed–load characteristic curve.

**Table 1 micromachines-14-00954-t001:** Material constants.

Material	PZT-5H	Al7075	ZrO_2_
Density(kg/m^3^)	7500	2810	6020
Poisson’s ratio	0.3	0.33	0.25
Elastic modulus(×10^10^ N/m^2^)	$\left(\begin{array}{cccccc} 12.72 & 8.02 & 8.47 & 0 & 0 & 0\\ 8.02 & 12.72 & 8.47 & 0 & 0 & 0\\ 8.47 & 8.47 & 11.74 & 0 & 0 & 0\\ 0 & 0 & 0 & 2.29 & 0 & 0\\ 0 & 0 & 0 & 0 & 2.29 & 0\\ 0 & 0 & 0 & 0 & 0 & 2.35 \end{array}\right)$	7.2	32
Piezoelectric constant(C/m^2^)	$\left(\begin{array}{ccc} 0 & 0 & -2.74\\ 0 & 0 & -2.74\\ 0 & 0 & 5.93\\ 0 & 7.41 & 0\\ 7.41 & 0 & 0\\ 0 & 0 & 0 \end{array}\right)$	/	/
Dielectric constant	$\left(\begin{array}{ccc} 1704.4 & 0 & 0\\ 0 & 1704.4 & 0\\ 0 & 0 & 1433.6 \end{array}\right)$	/	/

**Table 2 micromachines-14-00954-t002:** Structure parameters of a single leaf-spring (unit: mm).

Parameter	a	$a_{1}$	*L*	*L* _1_	b
Value	1.2	2	23	3	5

**Table 3 micromachines-14-00954-t003:** Results of the different calculation methods.

Methods	$U_{x}$ (μm)	$U_{y}$ (μm)
Theoretic analysis	48.05	1.39
FEM	48.46	1.23
Experiment	47.65	1.6

**Table 4 micromachines-14-00954-t004:** Comparison between speeds of some previous piezo inertia actuators and those in this work.

Literature	[15]	[16]	[17,18]	[19]	This Work
Driving frequency(Hz)	2000	1000	700	3500	800
Driving voltage(V)	100	100	100	100	100
Maximum Speed(mm/s)	14.25	7.12	18.37	0.7	27.077
Resolution(μm)	0.04	0.01	0.198	0.04	0.0325
Load(g)	350	158	270	4000	200
Transverse displacement(μm)	15.95	20.67	27.27	22	47.65
Vertical displacement(μm)	10	10	/	11	1.6
Displacement ratio (%)	62.7	48.3	/	50	3.4

## Data Availability

Data underlying the results presented in this paper are not publicly available at this time but may be obtained from the authors upon reasonable request.

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
