# Peer review of "A Novel Piezo Inertia Actuator Utilizing the Transverse Motion of Two Parallel Leaf-Springs"

_micromachines, 2023, doi:10.3390/mi14050954_

Round 1

Reviewer 1 Report

Dear Authors,

My comments are listed in the attachment.

Kind Regards

Dear Authors,

The quality of the language is fine. 

Kind Regards

Author Response

Thank you for giving us the chance to make our manuscript better. We have revised the main content of the manuscript according to the comments carefully. Revised parts of the manuscript are marked in red. At the same time, we have improved the writing of the paper. Our point-by-point response to the comments is as follows. Please check the attachment.

Reviewer 2 Report

1. The authors should provied more discussions on the presented data,could  more leaf-springs be used,  three or more ?

2. The figures should be reorganized, pls provide the standard deviation of Figure 8 and Figure 11b.

3. As for the funding, what is 'a Specialized Research Fund' means? 

Moderate editing of English language is needed.

Author Response

Thank you for giving us the chance to make our manuscript better. We have revised the manuscript according to the comments carefully and we hope the manuscript will meet the requirement of the magazine. Revised parts of the manuscript are marked in red. At the same time, we have improved the writing of the paper. Our point-by-point response to the comments is as follows. Please check the attachment.

Reviewer 3 Report

This study proposes a novel linear piezo inertia actuator based on the transverse motion principle. The idea is of interest. The main contribution of this study is to introduce the rectangle flexure hinge mechanism with two parallel leaf-springs. Therefore, the transverse motion of two parallel leaf-springs can be excited by using the piezo-stack nested in the flexure hinge mechanism, resulting in simplifying the actuator structure and improving output characteristics. This paper structure is very complete, including simulation results and experimental section. However, several comments on this manuscript are suggested as follows.

1)          The structure is innovative. There are a lot of detailed mistakes in the manuscript thus the authors are recommended to carefully edit the paper to avoid sloppiness. Some of those mistakes are listed: 1) there should always be a space between a number and the units. Some errors in the text; 2) the sentence “The actuator prototype is shown in Figure 5(b) and the size is 60×60×13.5 mm”, there is a unit error in this sentence; 3) the two displacement symbols in Table 3 are different in size.

2)          The sentence has a slight error: “Here, the angle was 10 and the push force F was 200N.” The tense of the sentence should correspond to the text and should be the general present tense.

3)          According to the main text, the ‘Y -direction’ and ‘Z-direction’ in the figure caption of Fig. 3 should be ‘y-direction’ and ‘z-direction’.

4)          The author should explain the reason for the deviation between the experimental value and the theoretical analysis value in Table 3.

ok

Author Response

Thank you for giving us the chance to make our manuscript better.  We have revised the manuscript according to the comments carefully and we hope the manuscript will meet the requirement of the magazine. Revised parts of the manuscript are marked in red. Please check the attachment.

Round 2

Reviewer 1 Report

Dear Authors,

I do not have any other comments related to the revised version of the paper. The manuscript can be published after minor revison (methodological errors and text editing)

Kind Regards

English is fine